# No Influence of Previous *Coxiella burnetii* Infection on ICU Admission and Mortality in Emergency Department Patients Infected with SARS-CoV-2

**DOI:** 10.3390/jcm11030526

**Published:** 2022-01-20

**Authors:** Jesper M. Weehuizen, Rik van Spronsen, Andy I. M. Hoepelman, Chantal P. Bleeker-Rovers, Jan Jelrik Oosterheert, Peter C. Wever

**Affiliations:** 1Department of Internal Medicine and Infectious Diseases, University Medical Center Utrecht, 3584 CX Utrecht, The Netherlands; rikvanspronsen123@gmail.com (R.v.S.); I.M.Hoepelman-2@umcutrecht.nl (A.I.M.H.); J.J.Oosterheert@umcutrecht.nl (J.J.O.); 2Department of Internal Medicine and Infectious Diseases, Radboud Expertise Center for Q Fever, Radboud University Medical Center, 6525 GA Nijmegen, The Netherlands; Chantal.Bleeker-Rovers@radboudumc.nl; 3Department of Medical Microbiology and Infection Control, Jeroen Bosch Hospital, 5223 GZ ‘s-Hertogenbosch, The Netherlands; P.Wever@jbz.nl

**Keywords:** *Coxiella burnetii*, Q fever, SARS-CoV-2, COVID-19, Dutch Q fever outbreak

## Abstract

Background: the geographical similarities of the Dutch 2007–2010 Q fever outbreak and the start of the 2020 coronavirus disease 19 (COVID-19) outbreak in the Netherlands raised questions and provided a unique opportunity to study an association between *Coxiella burnetii* infection and the outcome following SARS-CoV-2 infection. Methods: We performed a retrospective cohort study in two Dutch hospitals. We assessed evidence of previous *C. burnetii* infection in COVID-19 patients diagnosed at the ED during the first COVID-19 wave and compared a combined outcome of in-hospital mortality and intensive care unit (ICU) admission using adjusted odds ratios (OR). Results: In total, 629 patients were included with a mean age of 68.0 years. Evidence of previous *C. burnetii* infection was found in 117 patients (18.6%). The combined primary outcome occurred in 40.2% and 40.4% of patients with and without evidence of previous *C. burnetii* infection respectively (adjusted OR of 0.926 (95% CI 0.605–1.416)). The adjusted OR of the secondary outcomes in-hospital mortality, ICU-admission and regular ward admission did not show an association either. Conclusion: no influence of previous *C. burnetii* infection on the risk of ICU admission and/or mortality for patients with COVID-19 presenting at the ED was observed.

## 1. Introduction

Q fever is a zoonosis, caused by the intracellular Gram-negative bacterium *Coxiella burnetii.* Infection occurs through inhalation of contaminated aerosols, derived from sheep, goats and cattle. The bacterium is very resistant to death by environmental factors and can be spread by the wind. Person-to-person transmission occurs rarely. In 60% of humans, infection results in acute Q fever, an influenza-like illness, sometimes accompanied with mild to severe pneumonia or hepatitis [1]. The Dutch Q fever outbreak occurred between 2007–2010 and was globally unique for its size and duration. Seroprevalence screening for *C. burnetii* revealed that around 40,000 people were infected around the epicenter of the outbreak [2]. Most infections were reported in the south-east of the Netherlands, due to a high density of goats [2,3]. In the years after the epidemic, focus shifted towards patients with long term effects of Q fever suffering from Q fever fatigue syndrome (QFS) or chronic Q fever [4,5].

On 27 February 2020, the first patient in the Netherlands tested positive for the new severe acute respiratory syndrome coronavirus 2 (SARS-CoV-2) [6]. As the virus spread, the southern region of the Netherlands quickly became the epicenter of the Dutch coronavirus disease 2019 (COVID-19) outbreak [7]. Soon after, voices emerged that the epicenter of the COVID-19 outbreak showed striking geographical similarities with the epicenter of the Dutch Q fever outbreak. (Figure 1)

Immune dysregulation is linked to adverse clinical outcomes in COVID-19 patients [8]. The association between age, chronic diseases (e.g., diabetes mellitus and obesity) and severe SARS-CoV-2 infection are partially explained by immune dysregulation [8,9,10,11]. In addition, chronic or past infections, e.g., infections with Hepatitis B virus (HBV) and cytomegalovirus (CMV), are associated with immune dysregulation and potential adverse clinical outcomes in COVID-19 patients [12,13]. Earlier studies showed altered cytokine responses to viral ligands and lipopolysaccharide in Q fever patients suffering from QFS. The mRNA expression of pro-inflammatory cytokines interleukin (IL)-1β and tumor necrosis factor (TNF)-α was decreased and an increased susceptibility to respiratory infections was suspected [14]. Another study demonstrated increased concentrations of anti-inflammatory cytokines IL-10 and IL-1Ra at six months after acute Q fever infection. It was suggested that this was the result of altered transcriptional programming of myeloid cells [15]. As a result of this inflammatory dysregulation, Q fever patients might be more susceptible to infections, e.g., with SARS-CoV-2, and may have an increased risk of symptomatic disease and hospitalization. On the other hand, it is also possible to hypothesize a protective effect after an infection with *C. burnetii*. It is suggested that induction of the innate immunity by Bacillus Calmette-Guerin (BCG) vaccination or previous infections could lead to protection against other (intracellular) pathogens [16,17]. BCG vaccination already showed protection against respiratory tract infections in elderly patients [18,19,20] and trials assessing the protection of BCG vaccination against SARS-CoV-2 are currently conducted [21,22,23].

The geographical similarities of the Q fever outbreak and the start of the COVID-19 epidemic in the Netherlands raises questions and provided a unique opportunity to study an association between *C. burnetii* infection and the outcome following SARS-CoV-2 infection. As a possible consequence of this association, previous *C. burnetii* infection could be added as risk factor for severe SARS-CoV-2 infection with the appropriate preventive and therapeutic measure for these patients. The aim of this study is to assess the possible presence of an association between a previous *C. burnetii* infection and adverse outcome following SARS-CoV-2 infection. This association was studied in patients infected with SARS-CoV-2 during the first COVID-19 wave who presented at the emergency department (ED) of two hospitals located in the region highly affected by both the 2007–2010 Q fever epidemic and the first wave of the COVID-19 pandemic.

## 2. Methods

### 2.1. Study Design

We performed a retrospective cohort study in two Dutch hospitals. We assessed if there was evidence of a previous *C. burnetii* infection in adult patients (≥ 18 years old) diagnosed with COVID-19, defined as a PCR-confirmed SARS-CoV-2 infection. Patients that were included had presented at the ED during the first COVID-19 wave between 23 February and 31 May 2020, in the Jeroen Bosch Hospital in ’s-Hertogenbosch or Bernhoven in Uden. The primary outcome was a combined outcome of in-hospital mortality and intensive care unit (ICU) admission. Secondary outcomes were in-hospital mortality, ICU admission, and admission to a regular ward. We chose for a combined primary outcome because ICU admission in itself was affected by restriction in ICU admission due to lack of free ICU beds during the first wave of the pandemic. The combined outcome reflects bad outcome in COVID-19 patients and is not affected by this restriction.

### 2.2. Evidence of Previous C. burnetii Infection

Evidence of previous *C. burnetii* infection was assessed in the included COVID-19 patients. First, the electronic health records (EHR) were consulted. A previous positive result of *C. burnetii* PCR on serum, plasma or tissue, or a positive *C. burnetii* IgM or IgG antibody result was defined as a previous *C. burnetii* infection. A previous negative serology result obtained after the Q fever pandemic, thus during or after 2011, was defined as no previous *C. burnetii* infection. If no information on previous *C. burnetii* infection could be obtained from the EHR, stored blood samples of these patients from regular clinical care or the “BioMarCo-19” study (BioMarCo-19–CMO 2020-6344) were assessed for *C. burnetii* antibody status. Phase I and Phase II IgG *C. burnetii* antibody titers were measured using indirect immunofluorescence assay (IFA) (Focus Diagnostics, Cypress, CA, USA). Titration was performed with binary serial dilutions, with a detection cut-off titer of 1:64.

### 2.3. Data Collection

Baseline characteristics, comorbidities including date of diagnosis, clinical data and outcome status of the included patients were obtained from the EHR. Comorbidities included cardiovascular disease, diabetes mellitus requiring medication, any (hematologic) malignancy, obstructive pulmonary diseases defined as asthma, chronic obstructive pulmonary disease (COPD), interstitial lung disease (ILD) or cystic fibrosis (CF) and immunocompromised state, defined as the use of corticosteroids (prednisone or equivalent, cumulative dose > 700 mg), anti-CD20 therapy, biologicals (TNF-alpha inhibitors, interleukin-5 inhibitors and monoclonal antibodies), methotrexate, azathioprine and/or mercaptopurine within the last six months, having received an autologous/allogenic stem-cell transplantation, having neutropenia (<0.5 × 10^9^/L), (functional) hypo/asplenia, CD4-penia (<200 cells/mm^3^), hypogammaglobinemia and/or having a primary immunodeficiency. Furthermore, the updated Charlson comorbidity index (uCCI) was calculated [24]. Upon admission at the ED, vital parameters were documented, which we used to determine the modified early warning score (MEWS) [25].

### 2.4. Statistical Analysis

Data analysis was performed with SPSS (IBM SPSS Statistics for Windows, version 26.0.0.1. IBM Corp., Armonk, New York, USA). Figures were generated in SPSS and Microsoft Excel (Microsoft office for windows, Version 2013, Microsoft, Redmond, WA, USA). Multiple imputations were used to account for missing data on patient height and weight. We used both determinants, potential confounders and outcome variables in the imputation model and missing values were imputed under the assumption of missingness at random. Differences between patients with or without evidence of previous *C. burnetii* infection were assessed by univariate analysis using a Pearson’s Chi-square test or Fisher’s exact test for differences in proportions of binary variables and Mann–Whitney U test or independent *t*-test for continuous variables, as appropriate. We compared the outcomes between patients with or without evidence of previous *C. burnetii* infection using adjusted odds ratios (OR) with a 95% confidence interval (CI) from multiple logistic regression. Age, sex and body mass index (BMI) were entered as confounders in the model. Comorbidities diagnosed before the end of the Q fever pandemic, before the start of 2011, were entered in the model if they were present in more than 10 patients.

## 3. Results

### 3.1. Study Population

During the first COVID-19 wave 1151 patients were diagnosed with COVID-19 in the Jeroen Bosch Hospital in ’s-Hertogenbosch and Bernhoven in Uden. From 486 patients (42.2%), no information on previous *C. burnetii* infection could be obtained (no previous *C. burnetii* PCR or serology results), and no stored blood samples were available. Of the remaining patients, 36 patients (3.1%) were diagnosed with COVID-19 by serology only. These patients were also excluded from the analysis. Thus, in total, 629 patients (54.6%) were included in this study (Figure 2). The number of patients with a positive SARS-CoV-2 PCR included per week is displayed in Figure 3.

### 3.2. Evidence of Previous C. burnetii Infection

In the included patients, 117 patients (18.6%) had evidence of previous *C. burnetii* infection while 512 patients (81.4%) had not (Figure 2). The information on previous *C. burnetii* infection was determined from earlier performed *C. burnetii* PCR and serology results extracted from the EHR in 133 patients (21.1%). In these 133 patients, 52 patients (39.1%) had evidence of previous *C. burnetii* infection while 81 patients (60.9%) had no such evidence. In 496 patients (78.9%), serology results were obtained through IFA performed on stored blood samples. In these patients, 65 (13.1%) had evidence of previous *C. burnetii* infection while in 431 patients (86.9%) serology showed no evidence of a previous infection.

### 3.3. Baseline Characteristics

Baseline characteristics are shown in Table 1. Mean age was 68.0 years (SD ± 12.5, with a mean BMI of 28.5 (SD ± 5.1). Cardiovascular disease, chronic lung disease and diabetes mellitus were the most noted comorbidities in the medical history. More than two-third of the study population needed oxygen suppletion at ED presentation and 435 patients (61.5%) received (hydroxy)chloroquine as COVID-19-specific treatment. The median MEWS at the ED was 2.0 (IQR 2.0–4.0).

### 3.4. Outcome

In total, 126 patients (20.0%) were admitted to the ICU and 156 COVID-19 patients (24.8%) died in the hospital. Overall, the combined primary outcome, admission to the ICU and/or in-hospital mortality, occurred in 254 patients (40.4%). Among patients with and without previous *C. burnetii* infection, the primary outcome occurred in 40.2% and 40.4% of cases, respectively (adjusted OR of 0.926 (95% CI 0.605–1.416)). A total of 578 patients (91.9%) were admitted to a regular ward (regardless of ICU admission at any point). Among patients with and without previous *C. burnetii* infection, the adjusted OR for the secondary outcomes in-hospital mortality (22.2% vs. 25.4%), ICU admission (24.8% vs. 19.7%) and regular ward admission (89.7% vs. 92.4%) were 0.825 (95% CI 0.488–1.393), 1.299 (95% CI 0.799–2.112) and 0.612 (95% CI 0.303–1.237), respectively (Table 2).

## 4. Discussion

We found no decreased or increased risk of mortality, ICU admission, and/or admission to a regular ward after previous *C. burnetii* infection in patients admitted to the ED with COVID-19. Only one other study has been published assessing a possible association between previous *C. burnetii* infection and COVID-19 in an area with high Q fever incidence. *C. burnetii* seroprevalence was assessed in 50 Dutch patients admitted to the hospital with COVID-19. A slightly increased seroprevalence of *C. burnetii* IgG of 16.0% was found in these patients compared to seroprevalence in blood donors (12.2%) [26], and patients with a vascular of cardiac heart valve surgery in that area (14.9%). *C. burnetii* IgG seroprevalence among COVID-19 patients that were admitted to the ICU or died, was 15.7% and 9.0%, respectively. No clear association was found between previous *C. burnetii* infection and an adverse outcome of COVID-19 infection [27].

In our study, 18.6% of the COVID-19 patients had evidence of previous *C. burnetii* infection. However, this percentage cannot be compared directly to seroprevalence studies [2,26]. The percentages in seroprevalence studies are based solely on *C. burnetii* serology results at a certain time point. In our study, the percentage of patients with evidence of previous *C. burnetii* infection is based on both seroprevalence, and on earlier performed *C. burnetii* serology and PCR results on diagnostic samples extracted from the EHR. However, long term follow-up of acute Q fever patients showed loss of seropositivity (1.1% to 23.4%) some years after Q fever infection [28,29,30,31]. Therefore, the percentage of patients with evidence of previous *C. burnetii* infection in our study can be a slight overestimation because of the patients included based upon extra information from the EHR, or it can be underestimation because of possible loss of seropositivity in the long time period between the end of the Dutch Q fever and the start of the COVID-19 outbreak, compared to the seroprevalence studies.

The association between adverse outcomes after COVID-19 and several other infectious diseases is studied. Although immune dysregulation is described in patients infected with both SARS-CoV-2 and HBV, a co-infection did not significantly affect the outcome of COVID-19 [12,32]. An association between past CMV infection and adverse outcomes after infection with SARS-CoV-2 is hypothesized through increase of inflammatory mediated cytokines in these patients [13,33], however, a direct association study has not been performed to this date. A systematic review in people living with HIV described increased susceptibility to SARS-CoV-2 infection and a higher mortality [34].

Our population consisted only of patients who had visited the ED, a population that does not represent the whole population of COVID-19 patients. Consequently, we could not identify a possible protective or adverse effect of previous *C. burnetii* on the course of SARS-CoV-2 infection prior to presentation at the ED. All conclusions only apply to COVID-19 patients, given they were referred to the ED. However, in view of the relative high percentage of patients with evidence of previous *C. burnetii* infection presented at the ED, it could suggest that previous *C. burnetii* infection causes an increased susceptibility to SARS-CoV-2 and/or increases the chance of referral to the ED with COVID-19.

In the case that previous *C. burnetii* infection plays no role in the course of COVID-19 after ED presentation, the question arises which factors might be responsible for the large outbreak in this region at the start of the Dutch epidemic. The southern region of the Netherlands was the part of the country that had the earliest spring break period in 2020 and many inhabitants went skiing in Italy where SARS-CoV-2 was already spreading. Hence, travel to this region was advised against short after. This holiday-period coincided with the Carnival festival, which is a popular regional yearly four-day festival in the southern region during which many inhabitants come together in city centers and bars. Possibly, introduction from Italy and spread through Carnival could be an explanation for the initial emergence of SARS-CoV-2 in this region. Another possible explanation for the high incidence of COVID-19 is the high number of livestock farms in this region. The air quality around livestock farms is associated with higher risk of respiratory tract infection [35,36,37].

Furthermore, in literature, diet and nutrition status is proposed as having a potential impact on COVID-19 outcomes [38]. However, this will probably play no small role in our study because all patients are from the same region and there is no reason to assume the diet is different between the group of patients with and without previous *C. burnetii* infection.

Strengths of our research are the high number of patients with evidence of previous *C. burnetii* infection, which makes this population ideal to study the association between Q fever and COVID-19. The analysis was corrected for potential confounders, taking in mind that comorbidities diagnosed after *C. burnetii* infection cannot be a confounder and should not be corrected for. The data collected from the EHR was complete apart from information on patient height or weight, which were handled by multiple imputations.

Our research harbors several limitations. Firstly, the design of our study was of a retrospective nature; this can potentially introduce unmeasured confounders. As all data had to be extracted from the EHR, the confounders were limited to the date available within in the EHR. Secondly, a loss of seropositivity after acute Q fever is mentioned earlier. This can lead to misclassification of the determinant because patients with a previous *C. burnetii* infection could be classified as without infection. Misclassification of the determinant possibly leads to underestimation of the possible effect of the association. However, identifying previous *C. burnetii* infection through the EHR lowers the risk of misclassification in this study. In addition, patients were excluded from the analysis when no information on evidence of a previous *C burnetii* infection could be determined from patient history and no blood sample was stored. It is possible that more blood samples were stored from patients who had adverse outcomes. Therefore, these patients could be ‘missing not at random’ and potentially induce selection bias. We performed two sensitivity analyses testing the robustness of the estimates found in our initial analysis. First, an analysis excluding patients with evidence of previous *C. burnetii* infection determined through analysis of stored blood samples. Second, an analysis excluding patients with evidence of previous *C. burnetii* infection determined from the EHR. This way the analysis is separately performed for patients with a ‘symptomatic’ *C. burnetii* infection and those with an ‘asymptomatic’ *C. burnetii* infection. These sensitivity analyses did not change our initial findings. Lastly, the confidence interval around the estimate of the primary outcome is fairly broad. With the current sample size and ratio between the two groups a significant difference in risk of the combined primary outcome of ±15% could be detected. A total sample size of 1280 and 5000 COVID-19 patients is needed to detect a difference in risk of the combined primary outcome of 10% and 5%, respectively. However, the differences in adverse outcomes after SARS-CoV-2 infection in patients with and without previous *C. burnetii* infection are so small that the detection of a clinically relevant difference with a larger sample size is not expected. In conclusion, we report no association between previous *C. burnetii* infection and risk of ICU admission and/or mortality for COVID-19 patients visiting the ED.

## Figures and Tables

**Figure 1 jcm-11-00526-f001:**
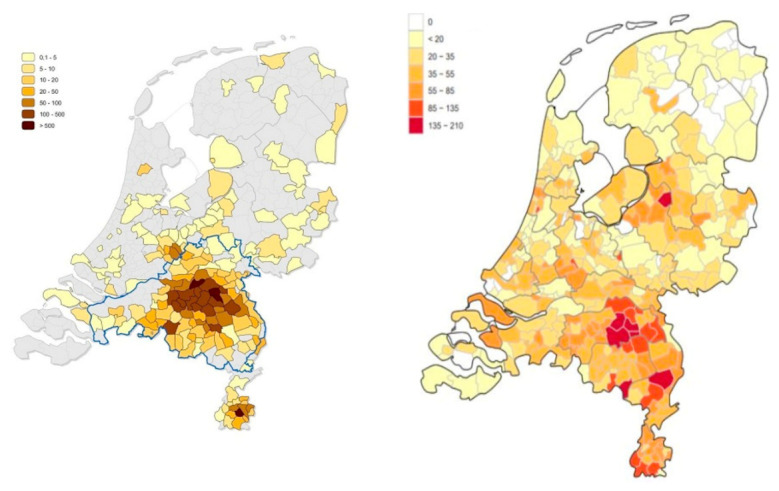
COVID-19 and Q fever in the Netherlands. Left: number of reported Q fever cases per 100,000 persons in 2009. Right: number of reported COVID-19 per 100,000 persons cases on 24 May 2020. Source: National Institute for Public Health and Environment (RIVM).

**Figure 2 jcm-11-00526-f002:**
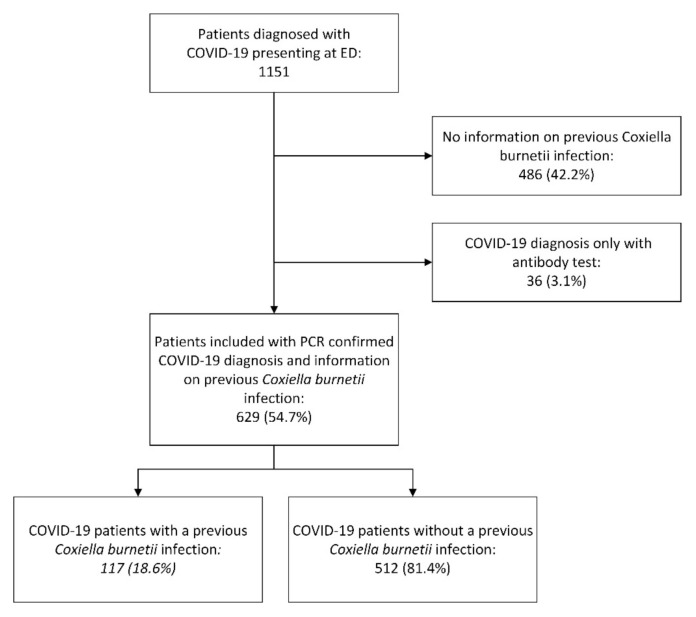
Flowchart of included patients. Abbreviations: COVID-19: coronavirus disease 19, ED: emergency department, PCR: polymerase chain reaction.

**Figure 3 jcm-11-00526-f003:**
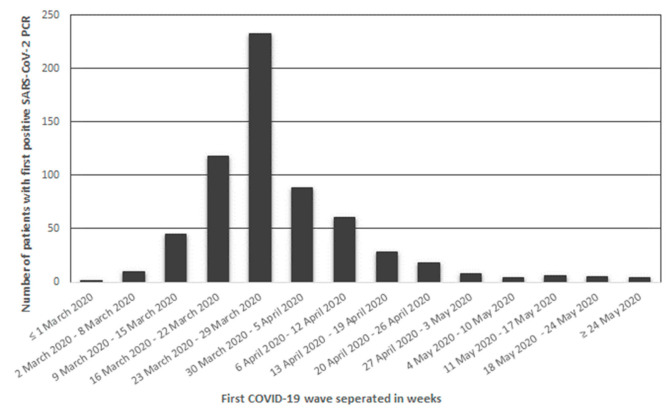
Histogram of number of patients presenting at the Emergency Departments of the Jeroen Bosch Hospital and Bernhoven with a first positive SARS-CoV-2 PCR per week during the first COVID-19 wave. Abbreviations: PCR: polymerase chain reaction, COVID-19: coronavirus disease 2019.

**Table 1 jcm-11-00526-t001:** Baseline characteristics ^a^.

Characteristic		Previous *Coxiella burnetii* Infection		*p*-Value ^b^
	Total *n* = 629	With *n* = 117 (18.6%)	Without *n* = 512 (81.4%)	
Age in years (mean, SD)	68.0 ± 12.5	67.46 ± 11.2	68.07 ± 12.7	0.635
Male sex (n, %)	412 (65.5%)	83 (70.9%)	329 (64.3%)	0.170
uCCI (median, IQR)	0.0 (0.0–1.0)	0.0 (0.0–1.0)	0.0 (0.0–1.0)	0.579
BMI (mean, SD)	28.5 ± 5.1	29.27 ± 5.0	28.35 ± 5.1	0.090
Medical history:				
Cardiovascular disease (n, %)	250 (39.7%)	52 (44.4%)	198 (38.7%)	0.250
Chronic lung disease (n, %)	111 (17.6%)	23 (19.7%)	88 (17.2%)	0.527
Diabetes mellitus (n, %)	141 (22.4%)	30 (25.6%)	111 (21.7%)	0.354
Immunocompromised state (n, %)	72 (11.4%)	9 (7.7%)	63 (12.3%)	0.123
Lymphoma (n, %)	9 (1.4%)	0 (-)	9 (1.8%)	0.222
Leukemia (n, %)	10 (1.6%)	4 (3.4%)	6 (1.2%)	0.157
Malignancy (n, %)	85 (13.5%)	18 (15.4%)	67 (13.1%)	0.512
With metastasis (n, %)	8 (2.5%)	2 (1.7%)	6 (1.2%)	0.646
Nursing home (n, %)	16 (2.5%)	3 (2.6%)	13 (2.5%)	1.000
ED presentation				
Oxygen suppletion (n, %)	435 (69.2%)	79 (67.5%)	356 (69.5%)	0.671
MEWS (median, IQR)	2.0 (2.0–4.0)	2.0 (2.0–4.0)	2.0 (1.0–4.0)	0.796
(Hydroxy)chloroquine treatment (n, %)	387 (61.5%)	75 (64.1%)	312 (60.9%)	0.526
Complaints at presentation				
Dyspnea (n, %)	441 (70.1%)	93 (79.5%)	348 (68.0%)	0.014
Cough (n, %)	411 (65.3%)	76 (65.0%)	335 (65.4%)	0.932
Thoracic pain (n, %)	59 (9.4%)	10 (8.5%)	49 (9.6%)	0.732

Abbreviations: n: number, SD: standard deviation, uCCI: updated Charlson comorbidity index, IQR: inter quartile range, BMI: body mass index, ED: emergency department, MEWS: modified early warning score. ^a^ Binary variables are presented as absolute numbers and percentages, continuous variables are presented as mean with standard deviation for normally distributed variables and median with interquartile range (IQR) for the non-normally distributed. ^b^
*p*-values were calculated using Pearson’s chi square test or Fisher’s Exact test to compare proportions between groups. An independent sample *t*-test was used to compare means for normally distributed continuous variables and a Mann–Whitney U test for non-normally distributed continuous variables.

**Table 2 jcm-11-00526-t002:** Risk of in-hospital mortality and ICU admission ^a^.

	Previous *Coxiella burnetii* Infection			
	With (*n* = 117)	Without (*n* = 512)	OR (95% CI)	Adjusted OR (95% CI) ^b^
Combined primary outcome	47 (40.2)	207 (40.4)	0.989 (0.657–1.490)	0.926 (0.605–1.416)
In-hospital mortality	26 (22.2)	130 (25.4)	0.840 (0.520–1.356)	0.825 (0.488–1.393)
ICU admission	29 (24.8)	101 (19.7)	1.341 (0.836–2.152)	1.299 (0.799–2.112)
Ward admission	105 (89.7)	473 (92.4)	0.721 (0.365–1.425)	0.612 (0.303–1.237)

Abbreviations: ICU: intensive care unit, OR: odds ratio, CI; confidence interval. ^a^ Variables are presented as absolute numbers and percentages. ^b^ Corrected for body mass index, sex, age, and immunocompromised state, a history of cardiovascular disease, chronic lung disease, diabetes mellitus, and a malignancy diagnosed before 2011.

## Data Availability

The data presented in this study are available on request from the corresponding author. The data are not publicly available due to privacy reasons.

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
