# Peer review of "No Influence of Previous Coxiella burnetii Infection on ICU Admission and Mortality in Emergency Department Patients Infected with SARS-CoV-2"

_jcm, 2022, doi:10.3390/jcm11030526_

Round 1

Reviewer 1 Report

The article by Weehuizen et al. is interesting because from an ecological observation, the higher rate of COVID-19 in an area where there was an outbreak of Coxiella, a study emerges that could shed light on mechanisms of susceptibility to COVID-19. The authors conducted solidly based research and concluded that there is no association. It is desirable to publish those well-designed articles that are not significant to avoid publication bias. The article needs to resolve the following issues.

MAJOR QUESTIONS

  • Can the authors verify that the Charlson Index (CCI) data in Table 1 are correct? It is striking that the mean score is 0 when the mean age is 68 years. In the CCI, two points are awarded when the age is in the range of 60-69 years, as the authors can see on the official CCI website https://www.mdcalc.com/charlson-comorbidity-index-cci. Another possibility would be that instead of the score, they have used the survival at ten years, but with a mean age of 68 years, it would be at least 90% or 0.90. The authors should review the table.
  • The study is not a retrospective cohort. It is a case-control study. Cases and controls are drawn from the reference population of people going to the emergency department with COVID-19. This study is a case-control study. Cases are the patients who have died, are in the ICU, hospital, etc. The controls are the patients who have not died, are not in the ICU, etc.

People in the outbreak area with and without Coxiella would have been selected in a retrospective cohort. THen it would have been checked if they had died, been in ICU, etc. This is not the case. On the other hand, relative risks are calculated in a cohort study, and odds ratios are calculated in a case and control. (As the authors have done). The authors should indicate that this study is a case-control study for all these reasons.

MINOR QUESTIONS

  • In the introduction, please indicate Coxiella's transmission mechanisms to allow an unfamiliar reader to understand the context. It would help to tell what hypothesis arose when you saw the similarity between the case distribution maps.

  • In Table 1, please put the % sign after the number when using a percentage to make it easier to read.
  • Finally, a question for discussion. Could diet have played a role ¿ Is the geographic area in which the Coxiella outbreak and covid cases occurred characterized by any particular dietary pattern? More than a thousand papers are studying the role of diet (as susceptibility) in the occurrence of Covid 19.

Reviewer 2 Report

The authors describe a really interesting, novel and well conducted observational study looking into the potential impact of a previous C. burnetii infection on the severity of outcomes in Covid-19 patients. Especially the discussion section is excellent and provides lots of useful insights. I only have a few minor suggestions which the authors might find helpful.

1) What was the rationale for using a combined primary outcome variable? This should be included in the manuscript.

2) In Table 1 it should be clarified which of the continuous variables are presented as mean with SD and which as median with IQR.

3) The authors might want to consider adding “ED” to the title (i.e. “No influence of previous Coxiella burnetii infection on ICU admission and mortality in ED patients infected with SARS-CoV-2”) as this is an important inclusion criterion for the study, as is rightly discussed (l. 233-240).

4) To address the concern regarding over-/underestimation of effect size in different subpopulations, a subgroup analysis of patients whose previous C. burnetii infection was identified from electronic health records vs patients whose stored blood samples were analysed specifically for this study might provide further insight. I’m not suggesting such an analysis is mandatory, but the authors might want to consider it.

5) What does “length” refer to (l. 126, l. 259)? Length of stay in ICU or hospital? Or, presumably, patient height?

6) The terms “sex” and “gender” appear to be used interchangeably. Maybe it is best to stick to one. Sex usually describes the biological aspect and gender the sociological component. I presume the focus in this study is on the former?

7) All calendar dates should be written correctly. “February 23th and May 31th” (l. 91) should be changed to “February 23rd and May 31st”. Similarly, dates in Figure 3 should be corrected.

8) Please use the point as decimal separator and the comma as thousands separator, e.g. write “18.6” rather than “18,6” (l. 24) and “40,000” rather than “40.000” (l. 38).

9) I would suggest changing “OR of 0.926” to “adjusted OR of 0.926” (l. 26, l. 189).

10) Please change “109” to “10^9” (l. 118) and “mm3” to “mm^3” (l. 119).

11) No need to write “MEWS-score” (l. 122, l. 183, Table 1) when the S in MEWS stands for score.

12) Consistent spelling should be used rather than alternating between “Q-fever” and “Q fever”.

13) There are a number of small typos and grammar mistake including “were didn’t show” (l. 27), “precious” instead of “previous” (l. 88), “an primary” (l. 119), “characteristics a” (l. 162), “admissiona” (l. 195), “IC” instead of “ICU” (l. 209), and “trough” (l. 229, l. 267).
